# Believing Processes during the COVID-19 Pandemic: A Qualitative Analysis

**DOI:** 10.3390/ijerph191911997

**Published:** 2022-09-22

**Authors:** Jolana Wagner-Skacel, Sophie Tietz, Eva Fleischmann, Frederike T. Fellendorf, Susanne A. Bengesser, Melanie Lenger, Eva Z. Reininghaus, Marco Mairinger, Christof Körner, Christoph Pieh, Rüdiger J. Seitz, Hannes Hick, Hans-Ferdinand Angel, Nina Dalkner

**Affiliations:** 1Department of Medical Psychology and Psychotherapy, Medical University Graz, 8036 Graz, Austria; 2Clinical Department of Psychiatry and Psychotherapeutic Medicine, Medical University Graz, 8036 Graz, Austria; 3Department of Psychology, University of Graz, 8010 Graz, Austria; 4Department of Psychosomatic Medicine and Psychotherapy, University of Continuous Education Krems, 3500 Krems, Austria; 5Department of Neurology, Centre of Neurology and Neuropsychiatry, Medical Faculty, LVR-Klinikum Düsseldorf, Heinrich-Heine-University Düsseldorf, 40625 Düsseldorf, Germany; 6Institute of Machine Components and Methods of Development, University of Technology Graz, 8010 Graz, Austria; 7Department of Catechetics and Religious Education, University of Graz, 8010 Graz, Austria

**Keywords:** COVID-19, cognition, emotion, credition, psychosocial functioning

## Abstract

Cognition, emotion, emotional regulation, and believing play a special role in psychosocial functioning, especially in times of crisis. So far, little is known about the process of believing during the COVID-19 pandemic. The aim of this study was to examine the process of believing (using the Model of Credition) and the associated psychosocial strain/stress during the first lockdown in the COVID-19 pandemic. An online survey via LimeSurvey was conducted using the Brief Symptom Inventory-18 (BSI-18), the Pittsburgh Sleep Quality Index (PSQI), and a dedicated Believing Questionnaire, which assesses four parameters of credition (propositions, certainty, emotion, mightiness) between April and June, 2020, in Austria. In total, *n* = 156 mentally healthy participants completed all questionnaires. Negative credition parameters were associated with higher global symptom load (from BSI-18): narratives: *r* = 0.29, *p* < 0.001; emotions *r* = 0.39, *p* < 0.001. These findings underline the importance of credition as a link between cognition and emotion and their impact on psychosocial functioning and stress regulation in implementing novel strategies to promote mental health.

## 1. Introduction

The ongoing coronavirus disease 2019 (COVID-19) pandemic is emotionally challenging for everyone; it shakes the world and modifies our ethics [1]. A wide variety of stress-related symptoms (including sleep disorders, depression, somatization, anxiety, and increased alcohol consumption) are potential consequences in large parts of the population worldwide, especially for individuals who were already vulnerable [2]. Furthermore, the outbreak of COVID-19 has posed challenges and great economic problems for the whole European continent with uncertainty about jobs and personal independence [3]. Supposed stressors were fear of infection, fear for relatives, fear of job loss, but also boredom and isolation. Stress has been considered as a physiological and behavioral response to a stimulus with adaptation to external demands [4]. The amount of perceived stress symptoms depends on personality structure, individual coping-mechanism, resilience, and other protective and non-protective factors [5]. Coping strategies with impact on stress related mental and physical health are defined as an action-oriented intrapsychic effort to manage stressful situations including individual differences [6]. Increased symptoms of general anxiety, depression, and distress were reported by younger people and especially females during the pandemic [7]. Psychological variables such as emotional stability and higher levels of dispositional self-control were found to be an important protective factor against perceived stress during the COVID-19 pandemic. The importance of personal and interpersonal skills as conscientiousness and agreeableness with the ability to remain calm and maintain emotional balance with a sense of acceptance could be useful to reduce stress [8]. Decisions on quarantine arrangements and social distancing measures could have had a further negative impact on psychological well-being and the attitude towards the COVID-19 pandemic. Brooks et al. investigated the effects of quarantine and observed associations with frustration, boredom, and post-traumatic stress symptoms [9]. There are numerous studies that have examined mental health symptoms as a response to the pandemic [10,11], as well as studies that have examined specific beliefs during the COVID-19 pandemic [12]. However, there is only one study from our research group [13] that investigates underlying believing processes.

Believing—the capacity to make a meaning out of others’ and one’s own behavior in terms of held mental states—is a highly developed human social and psychological achievement [14]. It involves a complex and demanding spectrum of capacities that are susceptible to different strengths, weaknesses, and failings.

Mental imagery has been maintained to be the basis of beliefs combining cognitive dimensions, culture, and social interactions [15] interwoven with emotional processes which are linked to cognitive operations and reflective awareness [16,17]. Importantly, personal beliefs might be challenged and even modified in such abnormal circumstances, as they are susceptible to new information contradicting earlier experiences [18]. In philosophy, belief is discussed as the state of mind, which might be associated with stability and stable beliefs [19]. In contrast, more recent approaches to belief place much more emphasis on its fluidity and character than on processes of belief [20,21].

The term “credition” is a neologism that is based on the Latin verb “credere” (to believe). It was coined to denote the fluidity and functionality of believing processes, and to indicate that beliefs and belief formation are narrowly connected with emotions [22]. It has recently been proposed that processes of believing are higher cognitive brain functions, which have emerged together with the evolution of the brain [21,23,24]. From a metatheoretical perspective, credition represents the processes which underpin believing and might be understood as part of the big ensemble of cognitive processes and functions.

Thus, the processes of believing belong to the driving forces which are interrelated to many other processes, such as perceiving, learning, valuating, planning, or decision-making [25,26,27]. Owing to fast neural processing, the majority of these coherent constructs is acquired typically without conscious awareness, whereas coherent constructs with language-based conceptual content become manifest as explicit beliefs, such as in autobiographic, religious, and political beliefs [25].

According to the Model of Credition, believing is not possible without an associated emotion. Thus, the model refers to a triadic substructure of psychological processes integrating cognition and emotion and assumes four parameters relevant for the believing process: propositions, certainty, emotion and mightiness [21].

A theoretically important issue is to clarify the relation between basic findings and their conceptual relevance for understanding believing on the one hand, and the design of the model on the other hand. Here, a continuing valuation is needed to adapt the model to innovative findings. The Model of Credition is a functional model. It describes the inner processes that take place between the beginning of the believing process at a certain instant in time and the end of the believing process at a certain instant in time. To understand the functionality, a translation of everyday believing situations into a model-related terminology is required. What might be called in our daily language “I believe” must be translated into a non-personal functionality which is “can be included”. For instance, if one says, “I believe the meat is eatable”, the important message to believe or to disbelieve is “eatable”. 

**Proposition**: In our example, “eatable” is the propositional content of a bab/clum. Bab is a neologism that was coined as a central unit of the functional process model of credition. As the believing process partly remains subliminal, it is important to integrate the aspect of subconsciousness. Blob and clum is the term which designates a subconscious bab. Thus, in the terminology of the model of credition, the mental mindset of someone is forced to start a believing process is called bab–blob configuration.**Certainty**: Regarding doubtable issues, persons may differ whether they are sure about them or not. In the language of the model of credition, we call this degree of certainty. In our example, the degree of certainty may differ between different persons. If someone has a degree of certainty of, for instance, 90%, they will be more ready to accept the clum “eatable” in their bab configuration than another person whose degree of certainty is only 55%.**Emotional loading**: The emotional loading of the clum eatable may be “disgusting”. In this case, it will be less likely that the clum can be integrated (i.e., believing that the meat is eatable) than in the case that the emotional loading might be shaped by “a little bit strange but interesting”.**Sense of mightiness**: The perspective of a subject is not limited to the emotions of a bab. It also includes the intensity of the emotion reflected by the sense of mightiness. If the emotional loading of disgust is mighty, it is less likely that the meat will be eaten than in the case that the disgust is more temporary and of a lower intensity.

For our study, we refer exclusively to the Model of Credition as it is published by Angel and Seitz [21] focussing on the three main elements credition, cognition, emotion, and providing the basis for a practical approach. This study aimed at investigating believing processes based on the Model of Credition and associations with psychosocial symptoms in mentally healthy people at the beginning of the COVID-19 pandemic. The uncertainty combined with emotional and cognitive modifications and individual abilities as personality traits lead to high levels of psychological stress in these times [28]. Individual differences in believing processes with cognitive and emotional responses to the lockdown and social distancing measures might be reflected in a worsened psychological response to the pandemic. Thus, we conducted a single-institution prospective mixed-methods analysis to address the question of how beliefs are formed and modified in times of deep crisis and what psychosocial impact this entails. Within this framework, the aim of this study was to answer the following exploratory scientific questions.

How do individuals experience the COVID-19 outbreak and which believing processes can be found?
Which credition profiles can be generated (proposition, certainty, emotional loading, sense of mightiness)?Is the propositional content related to the emotion and vice versa? What is the association between beliefs and psychological symptoms (depression, anxiety, somatization, global symptom load, and sleep quality)?

We hypothesized that propositional content (positive, negative, indifferent) and emotional loading would be associated with psychological symptoms.

## 2. Materials and Methods

This study took place during the first Austrian lockdown, which started on 16 March 2020, and encompassed travel restrictions, physical distancing, and the closure of institutions, such as schools, leisure venues, and nonessential shops. In April 2020, 2237 cases of COVID-19 infection were confirmed and 274 had died in relation to this disease. On 1 May 2020, strict measures were starting to get loosened: facilities were reopened, with shops, hairdressers, and leisure venues being opened first, and schools, restaurants, and places of worship following in the middle of May [29].

### 2.1. Procedure

The online survey was sent out via a survey tool (LimeSurvey 3.27.4 accessed on 1 June 2020) (between 9 April and 4 June 2020. Before participation, participants gave informed consent and answered the questionnaires anonymously or pseudo-anonymously (if they had a participant code from their participation in previous studies at the department). This study was part of the ongoing study “Psychological impact and effect of the corona virus (SARS-CoV-2) pandemic in individuals with psychiatric disorders—an online survey” (EK number 32-363 ex 19/20) at the Department of Psychiatry and Psychotherapeutic Medicine and was approved by the local ethics committee in accordance with the current revision of the Declaration of Helsinki, ICH guideline for Good Clinical Practice and current regulations (Medical University of Graz, Austria). Some parts of this study targeting psychiatric disorders with partially overlapping subjects have already been published [30]. 

### 2.2. Participants

The participants were recruited via social media or the LimeSurvey link was sent to relatives and acquaintances of the study group, or former study participants of the University clinic for Psychiatry and Psychotherapeutic Medicine. Included in this analysis were mentally healthy, German speaking adults who reported not having a psychiatric disorder, which was checked by two control items: 1. Do you have a diagnosed psychiatric disorder? (yes/no), 2. Do you have first-degree relatives with a severe mental disorder (schizophrenia, bipolar disorder, major depressive disorder? (yes/no). Participants were aged between 18 and 90 years, and gave informed consent. In total, 199 individuals were recruited. After excluding those not meeting the inclusion criteria, *n* = 156 remained for the final analysis. The sample partially overlaps with the subjects of the study by Tietz et al. who have surveyed believing processes in mental illness vs. healthy controls and have matched 52 individuals from this sample to their bipolar disorder sample [13].

### 2.3. Psychological Inventories

The following self-report questionnaires in German language were used in this study:

The Brief Symptom Inventory-18 (BSI-18) [31] was constructed by Derogatis and Fitzpatrick, based on the longer Symptom-Checklist-90-Revised (SCL-90-R) [32]. It was used to measure psychological symptoms during the last week (on a scale from 0–4), encompassing the Global Severity Index (GSI) as a sum of the three subscales Anxiety [Nervousness or shakiness inside, Feeling tense or keyed up, Suddenly scared for no reason, Spells of terror or panic, Feeling so restless you couldn’t sit still, Feeling fearful], Depression [Feeling no interest in things, Feeling lonely, Feeling blue, Feelings of worthlessness, Feeling hopeless about the future, Thoughts of ending your life], and Somatization [Faintness or dizziness, Pains in heart or chest, Nausea or upset stomach, Trouble getting your breath, Numbness or tingling in parts of your body, Feeling weak in parts of your body], each of them having acceptable internal consistency (Cronbach’s alpha: GSI α = 0.79, anxiety α = 0.68, depression α = 0.79, and somatization α = 0.63) [33]. 

The Pittsburgh Sleep Quality Index (PSQI) is a self-rated questionnaire that assesses sleep quality and disturbances over a one-month time interval [34]. The questionnaire consists of 19 items [example item: *During the past month, how would you rate your sleep quality overall? Very good—fairly good—fairly bad—very bad*], which generate seven components: subjective sleep quality, sleep latency, sleep duration, habitual sleep efficiency, sleep disturbances, use of sleep medication, and daytime sleepiness. Each component scores from 0 (no difficulty) to 3 (severe difficulty). A total PSQI score (range 0–21) of more than 5 yielded a diagnostic sensitivity of 89.6% and specificity of 86.5% (kappa = 0.75, *p* ≤ 0.001) in distinguishing between good and poor sleepers, whereas higher scores indicate worse sleep quality. Acceptable measures of internal homogeneity, consistency (test-retest reliability), and validity for the PSQI were obtained [34]. 

The self-constructed Believing Questionnaire, assessed credition parameters, and beliefs about the COVID-19 pandemic. The BQ consisted of six questions:


*When I think of the current very special situation, I believe:*

*When I think about my body, I believe:*

*When I think about my mental/emotional situation, I believe:*

*When I think of the coronavirus disease (COVID-19), I believe:*

*When I think about the time in three months, I believe:*

*When I think about the time in six months, I believe:*


The BQ assessed the four credition parameters: propositional content (narrative), degree of certainty, emotional loading, and sense of mightiness. Certainty was rated on a scale from 0–100 [*On a scale from 0 (=not sure) to 100 (=very sure), how sure are you about your belief?*]. The emotional loading was assessed with an “Emotion Wheel”, which consisted of three concentric circles. The innermost circle showed the six basic emotions as described by Paul Ekman: fear, anger, joy, sadness, contempt, disgust, and surprise [35], and the individuals had to choose one predominant emotion [*Please name an emotion that best describes your state while you are believing*]. Furthermore, the intensity of the emotion (sense of mightiness) was rated [*On a scale from 0 (not at all) to 100 (very much), how strongly do you experience the emotion while believing?*]. The difference between Item 5 and item 6 refers to the concept that credition is a process and describes the inner processes that take place between the beginning of the believing process at a certain instant in time and the end of the believing process at a certain instant in time.

Certainty and mightiness were metric variables, and emotion was categorized into positive (happy), negative (sad, angry, anxious, disgusted), and indifferent (surprised) emotions. In addition, we evaluated whether the narrative was positive or negative and whether it matched the emotion (congruent) or not (incongruent).

### 2.4. Data Analysis and Statistical Methods

The qualitative data of the BQ were coded and processed using MAXQDA 2020 (VERBI GmbH, 2019) [36]. The data were coded positive, negative, and indifferent (neither positive nor negative) by two independent raters according to the valence of the narratives or emotions. In total, there were six different codes (positive narratives, negative narratives, indifferent narratives, positive emotions, negative emotions, indifferent emotions). The resulting interrater reliability was κ = 0.95, which can be considered satisfying. The analyses could therefore continue with one coded data set.

For the analysis with the Statistical Package for Social Sciences (SPSS version 26, IBM, Armonk, NY, USA), six new variables were created, consisting of the respective frequencies of the different codes. In addition, a variable was created that measures the frequency of incongruence between the valence of a person’s narrative and the named emotion.

Partial correlation analyses with age, sex, education, relationship status, children, and current occupation as control variables were used for the correlations between the credition parameters and the scores in the BSI-18 and PSQI.

Bonferroni correction was used to correct for multiple tests, with an adjusted alpha-level of 0.001. All data met the assumed criteria of variance and linearity. The criterion of normality was not met for all variables. According to the central limit theorem, the sample was adequately large (≥30) to assume a normal distribution. We used the software MAXQDA 2020 (VERBI Software, Berlin, Germany) for the qualitative analysis to present prepositions and emotions for each item of the study. Word clouds are a useful method to simultaneously visualize the words as well as their frequency. The following word clouds show the most frequently used words of each item translated from German into English, with a possible loss of information due to the translation.

## 3. Results

### 3.1. Descriptive Statistics

The final sample consisted of 156 participants (52 males, 104 females; mean age = 39.4 +/− SD). Descriptive statistics about BSI-18, PSQI, and credition parameters are reported in Table 1. Neither the participants nor their close contacts were tested positive for COVID-19 or were quarantined at the time of testing.

Table 2 depicts the frequencies of the coded credition parameters of the six BQ items in detail: narrative and emotion, both classified into the three categories positive, negative, and indifferent, as well as the incongruence of both variables. Positive narratives and emotions were most often expressed and showed the least incongruence when participants were asked about their current situation. In contrast, individuals reported negative narratives and emotions most frequently in relation to the coronavirus. The greatest incongruence between narratives and emotions was found when asking about the time in three months.

The descriptive statistics of the certainty and the sense of mightiness are shown in Table 3. Participants were most confident when narrating about their situation and the coronavirus, and least confident when thinking about the time in six months; however, all six BQ items displayed similar mean values and standard deviations. The same applies to emotional mightiness, which was rated the highest concerning the mental/emotional situation and the lowest regarding the time in six months.

### 3.2. Correlations between Psychological Variables and Credition Parameters

Correlations that remained significant after Bonferroni correction were found between psychological variables and both narratives as well as emotions with negative as well as positive connotations (see Table 4). Positive narratives correlated negatively with the BSI-18 Depression score. Negative narratives were positively associated with the BSI-18 score and the BSI-18 Depression score. Negative correlations were found between positive emotions and the BSI-18 score, the BSI-18 Somatization score and the BSI-18 Depression score. Negative emotions were positively related to the GSI score and the BSI-18 Depression score. No significant correlations could be found between any of the following variables: indifferent narratives, indifferent emotions, incongruence, certainty, mightiness, BSI-18 Anxiety, and PSQI score.

### 3.3. Word Clouds

Figure 1 displays the word clouds showing the most frequently used words of the study population concerning their beliefs (narratives) during the first lockdown of the COVID-19 pandemic along the six items of the BQ. The most frequently used words of each item were translated from German into English, with a possible loss of information due to the translation.

Word clouds for item 1 (“When I think of my current situation, I believe”) show that individuals used a total of 253 words with the most often used words “I”, “again”, “everything”, “optimistic”, “sanguine”, “m2”, “content”. For item 2 (“When I think about my body, I believe”), individuals used a total of 273 words and for item 3 (“When I think about my mental/emotional situation, I believe”) 292 words. For item 2, individuals used the word “I” most often, followed by positive emotional words as predominantly used positive words about their body, such as “fit”, “fitter”, and “healthy”. As for item 3 and item 4 (“When I think of the coronavirus, I believe”), the individuals used the word “I” most frequently in item 5 (“When I think about the time in three months, I believe”) followed by positive emotion words such as “content”, “good”, “balanced”, and “calm”. For item 6 (“When I think about the time in six months, I believe”), it was notable that individuals used the word “We” most frequently, followed by positive emotion words such as “optimistic”, “hopefully”, “normality”, and “sanguine”.

## 4. Discussion

In this study, creditions according to the Mode of Credition by Angel and Seitz [21] were investigated using a self-constructed questionnaire measuring reported processes of believing during the COVID-19 pandemic. The activation of believing processes in response to adaptive stress behaviors shows the use of positive, negative, and indifferent narratives as well as positive, negative, and indifferent emotions.

The concept of credition as functionally understood processes of believing and the derived framework of the Credition Model assume cognitive and emotional efforts to manage the specific internal and external demands during the first lockdown in Austria with the imaginative ability to act in one’s own terms of mental states. This is the first implementation of this model in an empirical context and has already brought successful data with mentally ill individuals in the study by Tietz et al. [13]. This goes in line with the study by Seitz, emphasizing the relevance of the concept in neuropsychological and neuropsychiatric disorders [25].

The findings suggest that positive narratives and emotions related to the subjective situation during the COVID-19 pandemic were correlated negatively with psychological distress parameters. Stress adaptation as the relationship between a person and the environment is appraised by the person as taxing or exceeding their individual cognitive and behavioral efforts to manage the specific demands [37]. Recent conceptualizations of coping behavior focus on a more flexible and situational use of positive and negative coping strategies [38]. Positive coping includes behavior such as the use of social support, problem solving, or the cognitive reappraisal of an individual’s psychological capacity to adapt to adverse environmental circumstances. Individual beliefs play an important role in this context and are relatively stable accounts of what a subject holds to be true and to predict future events [39]. Organisms have to act upon incomplete information and reward uncertainty in a changing environment [40,41]. 

It is important to consider the relationship between coping strategies and stress in recent literature. Bongelli et al. showed that emotionally focused coping was negatively related to perceived stress and dysfunctional coping was positively related to stress [28]. This emotionally focused coping under stressful situations may allow an accurate perception of one’s own mental state cautiously comparable to defense mechanism. According to psychodynamic therapy, defense mechanism with splitting, projection, and projective identification are three mechanisms of the ego with protection and coping strategies which fulfill the task of making or keeping unconscious unpleasurable/negative effects, including feelings of fear, pain, or guilt. This is only to be regarded as pathological if the defense processes lead to a significant reduction in free self-development and self-realization, as well as a restriction of the ego function [42]. This may mean that a defense mechanism was functional at a certain point in development, ensuring ego protection and accommodating the demand, but, over time, that same defense mechanism can do the opposite and become dysfunctional.

In this study, there was a strong correlation between positive or negative narratives with the symptoms of distress and depression. In addition, the other credition parameters, such as certainty, emotion, and mightiness were also associated with self-rated stress symptoms. 

Narratives as tertiary processes are linked to abstract cognitive operations and reflective awareness with language-based summarization of complex memory, cognitive, and executive functions, identity, and mindfulness [43]. Positive narratives, such as good, optimistic, and sanguine, are expressions of the idea of a progress narrative that we are heading for improvement. The future is thought of as a space full of possibilities that are open to design and controllable in a positive sense. In relation to the subject, this corresponds to self-growth and self-actualization, key concepts of positive psychology [44]. Foucault speaks of a security positive [45]. It is primarily no longer about self-improvement, but about self-protection. Negative narratives, such as insecure, excited, and disillusioned, carry the risk of normalizing negative expectations over a longer period of time. Believing processes with negative emotional loading including strategies such as aggression, escape, or avoidance, could expose the body and mind to sustained and increase allostatic load and therefore lead to elevated distress [46]. Thus, credition with negative emotional loading was associated with increased psychological symptoms, depression, and increased global distress in this study. The relationship between negative coping strategies and mental health is well documented [47]. The way in which individuals cope with stressful events is more significant to psychological and physical well-being than the frequency and severity of the stressful episodes themselves [37]. Belief formation is not only impaired in neuropsychological syndromes, but beliefs are in close relation to the pathophysiological etiology of distress and depressive symptoms [25]. 

Subjective probabilistic representations or beliefs are typically formed subconsciously as stable, non-verbal precursors, and possess high predictability for future actions [27]. Narratives that were written as an answer to the item “When I think about the time in 3 or 6 months, I believe” were, for example, in 3 months, “that everything will soon be back to normal” and in 6 months, “that we/I will have a drug against COVID-19”. Another person wrote about the time in 3 months “that absolute normality will not yet set in” and in 6 months “that it is better than now”. The world frequency analyses show that instead of the most frequent word “I” in the 1st, 2nd, and 3rd item, in the items 5 and 6 that are based on the beliefs for the future “I” changes to “We”, which could signalize that there is an inwardness inherent in one we. During the crisis, we are no isolated creatures, but people who inherent a soul or an idea that is uniting. With a “We”, we create joint action and determination. The most frequent words in the items were words like good, sanguine, and optimistic.

The trend to negative correlations between the credition parameter certainty and distress or depression underscores again the importance of confidence, self-esteem, and self-perception in the formation of distress and depression. Healthy features are characterized by a coherent sense of the self with self-perception and affective regulation with flexible functioning when stressed by external or internal conflicts. Incoherent sense of self, problems in self-other differentiation, and problems in affect and impulse regulation, mentalization, and inflexibility lead to rigidity in several domains [48]. The self with cognition and emotion has been investigated extensively in neuroscience and has been related to a cerebral network as well as resting state, enhanced perception, and embodied simulation [49]. The core self is considered as a trans-species functional entity based on subcortical midline structures in a mutually regulating process with a more complex reflective and conscious self-distributed system linked with external events [50]. Self-processing has been operationalized and associated with basic functions, such as perception, action, reward, and emotions [51]. Cognitive emotional regulation in distress or depression with maladaptive strategies, such as self-blame, rumination, and catastrophizing, or adaptive strategies, such as acceptance, positive refocusing on planning, and putting into perspective, were connected to psychopathology [52]. In the context of credition, these positive strategies with more certainty could be a protection against mood-related disorders. The certainty about the narrative and the emotion could lead to an imaginative and perceptive capacity to understand one’s own behavior and intention. Covering a wide range of intrapsychic and metacognitive processes, such as self-monitoring (cognitive awareness of oneself), mindfulness (emotional awareness of oneself), and theory of mind (understanding of beliefs), show that this certainty and congruence between emotions and narratives could be a strong, intentional, and multifaceted concept for action [53]. By considering these mental states, they can be experienced as subjective modulatory processes. Behavior can be perceived as the result of underlying emotions, thoughts, and beliefs that can be represented, integrated, changed, and regulated by actions or reappraisal [54]. Believing processes can therefore be viewed as a protective resource and the knowledge about credition may eventually be integrated into therapy plans.

### Limitations

There are several limitations that should be considered when interpreting the results: A major limitation is that we did not use a validated questionnaire to assess credition; thus, the Believing Questionnaire was self-constructed. Additionally, we cannot say for sure whether the questionnaire measures really those beliefs which are verbally expressed. The qualitative data had to be transferred into positive, negative, and indifferent categories with a reduction and loss of information. Future studies should use content analysis for qualitative distinction between positive, negative, indifferent emotions. Furthermore, the cross-sectional design does not allow for any causal conclusions. The surveys were sent out through individual emails, mailing lists, and social media platforms. There is no way of identifying, understanding, and describing the population that could have accessed and responded to the survey, and to whom the results of the survey can be generalized. Furthermore, online surveys are only completed by persons who are literate, motivated, and who have access to the internet, which further biases the results. The entire survey was quite long (at least 20 min with the BQ at the end) and participants may have had trouble concentrating until the end. 

## 5. Conclusions

This study suggests that believing parameters play a significant role in the processing of psychosocial stressors. The processes of believing, so-called creditions, appear to enable people to formulate and integrate their intentional strategies for a successful adaptation to adverse circumstances, such as the lockdown at the beginning of the COVID-19 pandemic. We suggest that thinking about and verbalizing beliefs is a helpful transtheoretical and transdiagnostic concept to focus the attention of participants to a so far little observed phenomenon and explain vulnerability for stress-related mental disorders and probably for their treatment. Additionally, this may indicate that early treatment of individuals with maladaptation to stress should focus on the association of cognition, emotion, and beliefs. Socially embedded treatment strategies with adaptive functioning of imaginations in relation to inter- and intrasubjective capabilities may foster the therapeutic outcome. 

## Figures and Tables

**Figure 1 ijerph-19-11997-f001:**
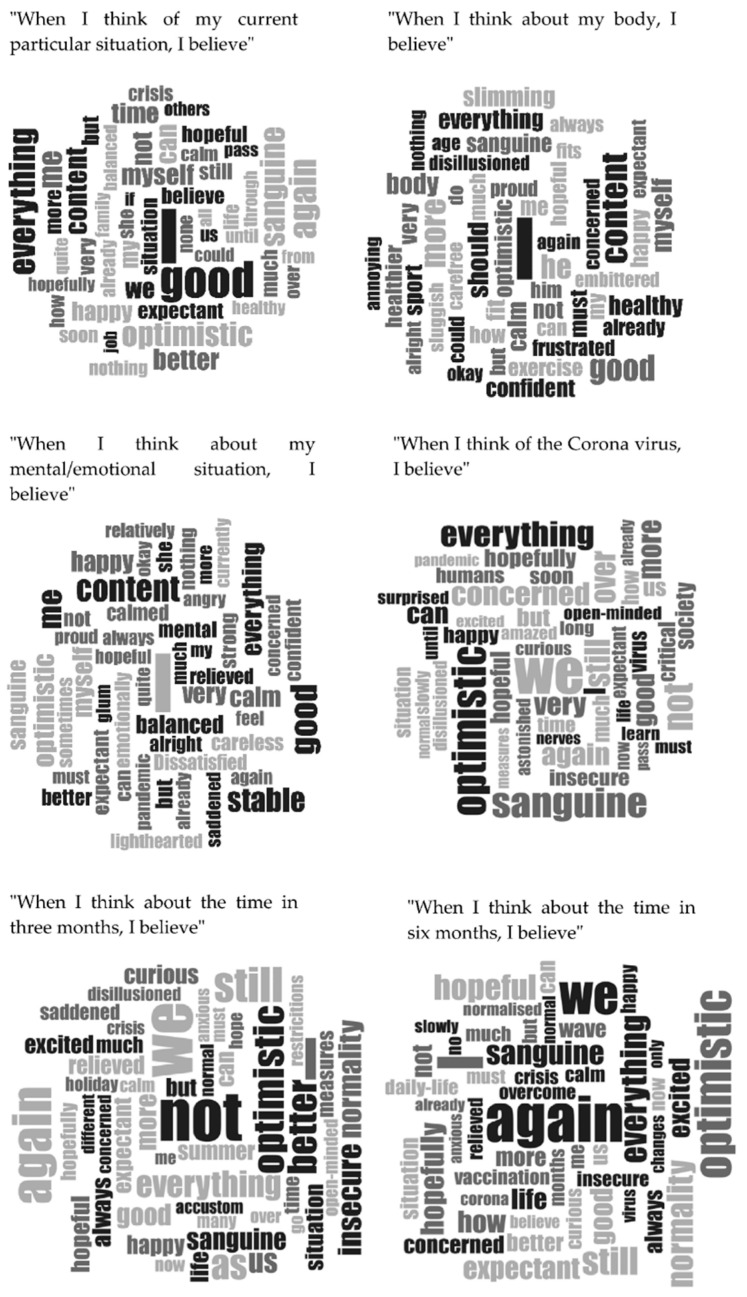
Most frequent words for the items of the Believing Questionnaire.

**Table 1 ijerph-19-11997-t001:** Descriptive statistics of psychiatric symptoms and credition parameters.

	*M*	*SD*	Min.	Max.
BSI-18 GSI	5.0	5.3	0	31
BSI-18 Somatization	1.1	1.8	0	12
BSI-18 Depression	2.1	2.7	0	17
BSI-18 Anxiety	1.8	2.1	0	11
PSQI	4.2	2.5	0	13
Positive narratives	3.7	1.6	0	6
Negative narratives	1.1	1.2	0	4
Indifferent narratives	1.2	1.1	0	5
Positive emotions	4.0	1.7	0	6
Negative emotions	1.7	1.6	0	6
Indifferent emotions	0.3	0.7	0	5
Incongruence ^1^	1.5	1.2	0	5
Certainty ^2^	82.6	12.2	25.7	100
Mightiness ^2^	75.3	15.0	26.8	100

M = Mean; SD = Standard deviation; BSI-18 = Brief-Symptom Inventory-18; GSI = Global Severity Index; PSQI = Pittsburgh Sleep Quality Index; ^1^ Incongruence between the narratives and the emotions; ^2^ in percent.

**Table 2 ijerph-19-11997-t002:** Frequencies of the credition parameters across the items of the Believing Questionnaire.

	Narrative						Emotion						Incongruence ^1^			
	Positive		Negative		Indifferent		Positive		Negative		Indifferent		Yes		No	
Item When I think …, I believe	*n*	%	*n*	%	*n*	%	*n*	%	*n*	%	*n*	%	*n*	%	*n*	%
…of my current particular situation…	118	75.6	17	10.9	21	13.5	130	83.3	23	14.7	3	1.9	30	19.2	126	80.8
…about my body…	91	58.3	24	15.2	41	26.3	104	66.7	46	29.5	6	3.8	46	29.5	110	70.5
…about my mental/emotional situation	113	72.4	24	15.4	19	12.2	122	78.2	31	19.6	3	1.9	27	17.3	129	82.7
..of the coronavirus disease (COVID-19)…	72	46.2	47	30.1	37	23.7	80	51.6	61	39.4	14	9.0	41	26.5	114	73.5
… about the time in three months …	86	55.1	36	23.1	34	21.8	88	56.4	53	34.0	15	9.6	49	31.4	107	68.6
… about the time in six months …	99	63.9	28	18.1	28	18.1	99	63.5	45	28.8	12	7.7	37	24.0	117	76.0

^1^ Incongruence between the narratives and the emotions.

**Table 3 ijerph-19-11997-t003:** Descriptive statistics of certainty and mightiness across the items in the Believing Questionnaire.

Item When I think …, I believe	Certainty				Mightiness			
	*M*	*SD*	Min	Max	*M*	*SD*	Min	Max
…of my current particular situation…	85.2	14.9	19	100	76.1	16.9	25	100
…about my body…	84.5	17.3	5	100	73.3	22.6	0	100
…about my mental/emotional situation	84.7	15.8	23	100	78.0	19.3	23	100
..of the coronavirus disease (COVID-19)…	85.6	15.4	23	100	76.3	18.9	14	100
… about the time in three months …	79.6	17.8	18	100	74.5	18.4	25	100
… about the time in six months …	75.7	19.3	15	100	73.5	20.2	15	100

**Table 4 ijerph-19-11997-t004:** Bonferroni-adjusted partial correlations between the credition parameters and the psychiatric symptomatology.

	BSI-18 GSI	BSI-18 Soma	BSI-18 Depr	BSI-18 Anxi	PSQI6
Positive narratives	−0.24 *	−0.17 *	−**0.28** ***	−0.10	−0.10
Negative narratives	**0.29** ***	0.17 *	**0.36** ***	0.13	0.14
Indifferent narratives	0.04	0.04	0.04	0.01	−0.03
Positive emotions	−**0.36** ***	−**0.29** ***	−**0.40** ***	−0.16	−0.17 *
Negative emotions	**0.39** ***	0.23 **	**0.45** ***	0.21 **	0.16
Indifferent emotions	−0.02	0.17 *	−0.07	−0.10	0.05
Incongruence ^1^	0.08	0.14	0.04	0.03	0.02
Certainty	−0.13	−0.07	−0.11	−0.13	−0.14
Mightiness	−0.06	−0.03	−0.02	−0.09	−0.04

In bold letters, the Bonferroni-corrected significant correlations (Bonferroni-adjusted *p* < 0.001); Influence of variables age, sex, education, relationship, children and current occupation is partialized out; BSI-18 = Brief-Symptom Inventory-18; GSI = Global Severity Index, BSI-18 subscales: Soma = Somatization, Depr = Depression, Anxi = Anxiety; PSQI = Pittsburgh Sleep Quality Index; ^1^ Incongruence between the narratives and the emotions; * *p* < 0.05; ** *p* < 0.01; *** *p* < 0.001.

## Data Availability

Not applicable.

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
