# Peer review of "Believing Processes during the COVID-19 Pandemic: A Qualitative Analysis"

_ijerph, 2022, doi:10.3390/ijerph191911997_

Round 1
Reviewer 1 Report
Thank you for the opportunity to review this paper.
The paper aims to investigate the role of the "Process of Believing" in influencing the psychological impact of the COVID-19 pandemic.
The aim is interesting and original.
Unfortunately, the methodology is frail, it does not always use standardized questionnaire and the qualitative analysis is very simple and not well explained.
I report below some suggestions. I think that the metholodogy should be rethought. The Introduction as well is not always clear and does not fit well, in coherence, with the method and results sections.
Abstract: you mean “BSI-18” instead of “GSI-18”? If they refer to the Global Severity Index, given that they do not talked before, they should indicate the name in the extended form.
Line 37-39: please provide a reference for this
Line 42-44: As in the sentence the authors also refer to personality factors and protective/nonprotective factors, I suggest to provide some more references. The reference they provide especially refer to coping mechanisms and resilince, not so much about personality. For instance: Flesia, L.; Monaro, M.; Mazza, C.; Fietta, V.; Colicino, E.; Segatto, B.; Roma, P. Predicting Perceived Stress Related to the Covid-19 Outbreak through Stable Psychological Traits and Machine Learning Models. J. Clin. Med. 2020, 9, 3350. https://doi.org/10.3390/jcm9103350
Line 80: “represents” instead of “represent”
Line 57-115: this section of the Introduction is not clear and it not helpful to understand the aim and background of the work. For instance, they talk about dopamin, oxytocin… That are aspects that the paper doesn’t deal with. At line 95-97 they say they will explain something about the model, however, what they explain does not fits well with the methodology of the work.
As in the research aims and research questions section, they talk about “proposition”, “certainty”, “emotional loading”, “sense of mightiness”, these are the concepts the authors should explain in the Introduction section, in order to explain the theoretical background about the “believing process”.
Line 144: what link? The authors did not talk about any link before
Line 146-149: it is not clear one aspect: all participant underwent the Structured Clinical Interview for DSM-IV, as part of the questionnaire?
Line 158: what do the authors mean with “pseudo-anonymously”?
Did the authors calculated the power analysis in order to determine the minimum sample required?
Line 167: therefore, in the inclusion criteria the authors could indicate that being German speaker was also an inclusion criteria.
Psychological inventories section: the authors should provide some example items for each questionnaire. They could also provide the complete questionnaire battery in the supplementary material, especially for self-constructed questionnaires (i.e. emotional distress and Believing Questionnaire).
Without this description it is difficult to understand results.
Table 2: I suggest to report the extended item, instead of 1,2,3,4,5,6
Line 247-248: how was certainty rated/measured?
Word clouds: how did the authors create the word clouds? They should also outline this issue in the Method section. Moreover, they should comment and clarify the figures in the Results section.
Line 316-322: this section of discussion seems me to be a little bit out of topic. In the paper the author did not use this categorization of emotions.
In the qualitative distinction between positive, negative, indifferent emotions, I would have found more relevant a content analysis, and some examples, with specific distinction about specific emotions.
Line 401: “appears” instead of “appear”
Author Response
See the attached document regarding the point-to-point response.

Reviewer 2 Report
The article is very interesting both because of the topic presented and because the authors argue that belief processes have important repercussions in psychological terms, which I certainly and fully agree with. I also found the introduction of a qualitative part particularly interesting. Nonetheless, it was not presented adequately. The labels positive, negative, indifferent are not sufficient. Some examples (which only appear in the discussions) should be added and discussed in the results section. However, both the section devoted to theoretical framework (Introduction) and the results section are weak and should be more clearly and thoroughly presented. References, for example, to coping strategies only appears in the discussion section. Furthermore, the cited bibliography is at least partly rather outdated and should be implemented in both the introduction and discussion sections. Research questions that are clearly presented in the Introduction section are not taken up in the discussions with the same level of clarity. The article could benefit from more clarity in this respect as well. The results should be better described.
Some detailed suggestions follow
Introduction
Lines 51-52: the authors claim that “there is only one study from our research group that investigates underlying believing processes [9]”, but this statement is too assertive and not exactly true. Numerous studies have been conducted on belief processes during COVID-19 pandemic. Just search google scholar. It is necessary to mitigate this statement and rephrase the gap, pointing out how this article on beliefs differs from others on the same topic.
Lines 69-1 and 82-92: several concepts are introduced in an excessively synthetic manner that should be better explained because they refer to very different disciplinary domains.
Lines 108-115 the reference model the authors claim to use should, already in this section, be made more explicit
Lines 118-121 “The uncertainty combined with emotional and cognitive modifications could lead to high levels of psychological stress in these times”. With respect to this statement, numerous bibliographical references can be cited. I suggest that the author read the article Bongelli, R., Canestrari, C., Fermani, A., Muzi, M., Riccioni, I., Bertolazzi, A., & Burro, R. (2021, August). Associations between Personality Traits, Intolerance of Uncertainty, Coping Strategies, and Stress in Italian Frontline and Non-Frontline HCWs during the COVID-19 Pandemic—A Multi-Group Path-Analysis. In Healthcare (Vol. 9, No. 8, p. 1086). MDPI and consider the bibliography cited there, particularly in the introductory section.
2.1. Participants
Lines 152-154: is the reference to the fact that a large part of the sample is the same as in another study really necessary?
Lines 158-159: what means “anonymously or pseudo-anonymously (if they have already participated in previous studies)?
2.2. Procedure
Lines 159-161: The authors claim that the study is part of a research study entitled 'Psychological impact and effect of the corona virus (SARS-CoV-2) pandemic in 160 individuals with psychiatric disorders - an online survey', but above, among the inclusion criteria the authors claim that “Inclusion criteria were healthy adults (no psychiatric disorder was previously diagnosed at the outpatient center for BD at the Medical University of Graz, Department of Psychiatry and Psychotherapeutic Medicine, using the Structured Clinical Interview for DSM-IV), between 18 and 90 years old”. The reader is confused.
Lines 184-205: it is all very confusing. The reader cannot understand what instruments were used; what questionnaires were administered. First the authors refer to 5 items and then to 6 questions; first they refer to a score from 0 to 4 and then 6 questions are presented that seem to be 6 open questions. The reader cannot follow the line of argument. Are these two different instruments? If so, as I imagine, they should certainly be better described.
3. Results
Due to the lack of clarity in the description of the instruments used, the data results are also difficult to follow. A better description of the former could also improve the latter.
Check that the signs (+ and -) of the correlated variables are all correct in Table 4
3.4. Word clouds
This paragraph needs to be rewritten. The reader does not understand anything at all. The clouds must be described.
Discussion
Lines 293-295: The author claims that they tested “the theory that the process of believing under stressful situations can allow a relatively accurate perception of one’s own mental states and reflecting 294 personal meaning with resulting adaptive orchestration and action”. Both the theory and the results, as already pointed out, should be better defined. In its current form, the article does not seem to achieve its stated objective.
Lines 310 ff.: The authors first refer generically to coping strategies, citing Bonanno et al. 2013 [32]. Firstly, I think the authors should have introduced coping processes in the introductory section as well; secondly, they should refer to the most recent literature on the relationships between COVID-19 and coping strategies (see, e.g., Bongelli et al. 2021and consult the bibliography presented there).
Lines 327-29: The author argues that “Narratives as tertiary processes are linked to 327 abstract cognitive operations and reflective awareness with language-based summarization of complex memory, cognitive, and executive functions, identity, and mindfulness”. Perhaps a figure showing these three processes would facilitate the readers' understanding.
Lines 349 ff.: For the first time, the authors report qualitative results that should have been presented in the results section. They should also have specified the questions (three months, six months...) in the methodology section. This missing information made the reading of the paper difficult and at times unintelligible, as noted above.
Author Response

(The authors gave the same response as above.)

Round 2
Reviewer 1 Report
The paper is now suitable for publication
Congratulations to the authors
Author Response
Dear Reviewers and Editor,
Dear Reviewer,
Thank you for your time and your review on our paper.
Attached you find the point-to-point response and the main changes are highlighted in the text marked yellow.

Reviewer 2 Report
Since the authors have certainly made the methodological part of their study more understandable and have accepted the requested suggestions, I believe it can be published.
I would only ask them to specify the meaning of the bab/clum pair because not everyone is familiar with the model of Runehov and Angel 2013.
Author Response
Dear Reviewer,
Thank you for your time and your review on our paper.
Attached you find the point-to-point response and the main changes are highlighted in the text marked yellow.
